# Links between Adolescents’ Engagement in Physical Activity and Their Attachment to Mothers, Fathers, and Peers

**DOI:** 10.3390/ijerph16050866

**Published:** 2019-03-09

**Authors:** Ausra Lisinskiene, Vida Juskeliene

**Affiliations:** 1Academy of Education, Vytautas Magnus University, 44248 Kaunas, Lithuania; 2Institute of Hygiene Lithuania, 01128 Vilnius, Lithuania; vida.juskeliene@hi.lt

**Keywords:** physical activity, attachment to parents, mother, father, adolescents, peers

## Abstract

Physical activity (PA) is one of the key components in promoting a healthy lifestyle in children. PA offers a number of health benefits to children and their families. However, a large proportion of children do not meet the current PA guidelines (at least 60 min of PA daily). The problem of insufficient PA could be explained in relation to early childhood when attachment between the child and the parent begins and family habits are formed. As a result, physical inactivity in adolescence is associated with negative health outcomes such as obesity, heart diseases, and cardiometabolic risk. Therefore, this study is aimed at examining the links between adolescents’ engagement in physical activity and their attachment to their mother, father, and peers (trust, communication, alienation) as well as their socio-economic status (SES). We applied a quantitative cross-sectional study design. A total of 835 students aged from 15 to 18 (females = 480 and males = 355, M age = 16.0, SD = 1.1) completed the questionnaire. This study revealed that physical activity had a weak positive correlation to mother (r = 0.13, *p* = 0.01) and father attachment (r = 0.18, *p* = 0.01), trust (r = 0.17, *p* = 0.01), and communication (r = 0.16, *p* = 0.01) with both parents and a weak negative correlation with father alienation (r = 0.13, *p* = 0.01). The overall study results show that adolescent communication to father, male gender, a younger age, and a higher SES are important factors in relation to adolescent physical activity.

## 1. Introduction

Although the benefits of physical activity have been well established, the physical inactivity of adolescents is increasing [1,2,3,4,5,6]. The majority of adolescents do not meet the WHO recommendation of moderate-to-vigorous physical activity [1,2,3,4,5] for at least 60 min daily. In Europe, participation in physical activities declines dramatically from age 9 to ages 11–15. Only 1 out of 5 adolescents regularly participate in physical activity [3,4,5]. Moreover, researchers investigated trends in physical activity from 2002 to 2010 across 32 countries in Europe and North America among 11–15 years adolescents. Despite the efforts of promoting physical activity to people in this age group, only a small increase in the proportion of adolescents meeting the recommendations [1] has been observed. However, these trends were not evident in all countries. Finland made the greatest improvement and Lithuania the greatest decline in rates of daily physical activity. The general decline in physical activity in Lithuanian schoolchildren between 1992 and 2012 has also been observed by Venckunas et al. [5]. Researchers have emphasized that, if this general negative trend continues, it will compromise the well-being of future adults and create a serious economic burden on society. Researchers highlight that not only the decreasing physical activity of adolescents but also a lack of motivation to take part in physical education classes pose the most serious problems. It is a relevant and growing problem that youths are facing nowadays. Parents, teachers, and other significant role models are increasingly being encouraged to incorporate physical activity into daily lifestyles of young adolescents [5].

As adolescent involvement in physical activity prepares adolescents for a physically active environment and promotes healthy lifestyle [6], there is a necessity to look for engaging, empowering methods of involving schoolchildren in physical education. It is important to determine how adolescents can be motivated to participate in physical activities and to encourage them to become involved in physical activities that are appropriate for their age and gender, that are enjoyable, and that offer a varying range of activity. There is a large body of literature supporting the notion that parents have the greatest influence on physical and sports activities [7,8] (p. 186); Attachment relationships would appear to be relevant to the motivation of physical and sports activity in adolescents [8,9]. Moreover, parental safety concerns have been recognized as a critical factor in adolescents’ physical activity [10]. Parents concerns are associated with parental decision-making and how much they encourage adolescents to exercise. Parents with greater safety concerns tend to be less likely to encourage or allow their children to participate in outdoor activities. Parents may also set time limits on physical activities, restrict adolescents’ range of activity (to areas close to home), and only allow activities that are conducted under adult supervision [11].

Although parental involvement has been identified as a crucial factor, the factors related to mother, father, and peer attachment on adolescent physical activity are still unclear. There is a difference between parent–child attachment and parental involvement. However, parental involvement will always depend on the level of parent–adolescent attachment. A phenomenon of parent–child attachment according to scientific studies is of the highest importance [8,12] and has a strong psychological bond within parent and child. Attachment between the parent and the child that is formed in early childhood is referred to as either secure or insecure [12]. Attachment theory states that the psychological and behavioral effects of early parent–child relationship will affect the development of close relationships with other people (for example with peers) in the future [12]. The basis for the development of attachment in children is the need for security, which can be provided by an adult person [12]. One of the important tenets of attachment theory is the notion that early childhood lays the foundations for the development of personality throughout one’s lifespan [12]. In contrast, parental involvement is often the result of the attachment. If the attachment between the parent and adolescent is insecure, the adolescent will likely reject the parent, and parental involvement will not be welcomed by the child or in other cases will not be offered to the adolescent by the parent. If the parent–adolescent attachment is secure and has a strong psychological bond brought from the early childhood, parental involvement will be welcomed by the adolescent and most likely parental involvement will be positive. The current study’s focus is on the psychological effects of an attachment to a mother, a father, and peers on adolescent physical activity, and scientific background into such insight is lacking. This study presents a unique insight as to how an adolescent lifestyle factor (physical activity) might be influenced by psychological relations in the background of adolescent–parent and adolescent–peer attachment. Moreover, with adolescent issues such as close friend acceptance, the quality of interpersonal relationships between the peers becomes particularly important [13,14,15] Researchers [13,14,15] have emphasized that adolescents’ attachment to parents or key caregivers are of utmost importance. However, it is still not clear how parents, especially mothers, fathers separately, and peers influence adolescent involvement in physical activities.

The study is aimed at examining the links between adolescents’ engagement in physical activity and their attachment relationships with their mother, father, and peers. We assessed whether the parent and peer attachment domains are associated with the level of physical activity among 15–18-year-old students from mainstream schools in Lithuania. Based on the existing evidence, we assumed that adolescents whose parents are supportive of physical activity are higher. We also assessed whether gender, age, and socio-economic status moderates these associations.

## 2. Materials and Methods

### 2.1. Participants

The participants were 835 adolescents (females = 480 and males = 355) between the ages of 15 and 18 years (M age = 16.0 SD = 1.1). In data analysis, the research subjects were divided into two age groups: middle adolescence (aged 15–16; *n* = 441) and late adolescence (aged 17–18; *n* = 394). The participants were recruited from the six mainstream schools from the three largest cities of Lithuania by applying a multistage sampling procedure.

### 2.2. Procedure

First, the ethical approval of the eligibility to conduct the research was obtained from the first author’s institution. Moreover, the researchers obtained permission from school administrators to conduct the study. Once the permission was granted, the researchers contacted the senior management of the target schools. Prior to the collection of data from the students, parental informed consent forms were also completed. Participation in the study was voluntary, and informed consent was obtained from every participant. Students completed the questionnaires with researchers present in the classroom.

### 2.3. Measures

Physical activity was assessed using the International Physical Activity Questionnaire for Adolescents (IPAQ-A) [16]. IPAQ-A is questionnaire appropriate for high school students (grades 9–12; approximately 15–19 years of age) who are currently in the school system. The self-administered questionnaire comprises eight items and collects information on participation in different types of activities and sports, effort during physical education classes, and activity during lunch, after school, evening, and at the weekend during the past 7 days. Mean composite score on a range of 1–5 was calculated (a score of 1 indicates low physical activity and a score of 5 indicates high physical activity). The final IPAQ-A activity summary score was obtained from the mean of the eight items. A higher score indicates higher levels of PA.

The inventory of parent and peer attachment (IPPA)—the mother, father, and peer version [17]—was used in this study. The IPPA was developed to assess adolescents’ perceptions of the positive and negative affective/cognitive dimension of relationships with their parents and close friends. Three dimensions are evaluated: degree of mutual trust; quality of communication; and extent of alienation. The trust subscale measures the degree of mutual understanding and respect in the attachment relationship, and a sample item is as follows: “My parents accept me as I am.” The communication subscale assesses the extent and quality of verbal communication, and a sample item is “My friends can tell when I’m upset about something.” Finally, the alienation subscale assesses feelings of anger and interpersonal alienation, and a sample item is “I feel angry with my parents.” The development samples were 16–20 years of age; however, the IPPA can be used with adolescents aged 12. The instrument is a self-report questionnaire with a five-point Likert-scale response format. The original version consists of 28 parents and 25 peer items, yielding two attachment scores. The revised version (the mother, father, and peer version) is comprised of 25 items in each of the mother, father, and peer sections, yielding three attachment scores. The IPPA is scored by reverse-scoring the negatively worded items and then summing the response values in each section [17]. The instrument has high internal consistency, with Cronbach’s alpha coefficients attachment domains for parent attachment; mother and father attachment domains range from 0.083 to 0.095, and peer attachment domains range from 0.062 to 0.090.

Finally, the socioeconomic status (SES) of the family [18] was measured using a self-reported item with the given alternatives: “I think that I and my family are: 1. significantly poorer than most Lithuanian people; 2. poorer than most Lithuanian people; 3. as wealthy as the majority of Lithuanian people do; 4. slightly wealthier than most Lithuanian people; 5. much wealthier than most Lithuanian people.” Mean score on a range of 1–5 was calculated. In the analysis, all responses were divided into two subcategories: “average or lower” (Categories 1–3) and “higher than average (Categories 4–5).

### 2.4. Data Analysis

All the analyses were performed using IBM SPSS Statistics for Windows software (version 22.0) (IBM, Chicago, IL, USA). All IPPA scales and IPAQ-A scale internal consistency was evaluated using Cronbach’s alpha statistic, where values close to 0.7 are considered good. All continuous variable distributions were checked with tests of normality to estimate whether parametric tests would be applicable. In order to determine whether physical activity and IPPA scales differ significantly between gender, age, and SES groups, independent sample T-tests were used together with Cohen’s *d* effect size calculations. According to [19], Cohen’s *d* effect size is interpreted as follows: 0.2 = small effect, 0.5 = medium effect, 0.8 = large effect. Characteristics of participants were described as means and standard deviations (SD). Pearson correlations were used to check for relationships between IPPA scales and selected demographics with physical activity. A multiple regression model using the enter method was developed to understand which variables and to what extent influence the physical activity of adolescents. Statistical significance was set at *p* < 0.05 for all tests.

## 3. Results

A random sample of 15–18-year-old pupils was used for the study. The sample size was calculated basing on the number of pupils in the population. For the sake of data precision, the sample size was calculated not by the total number of pupils in secondary schools of the Vilnius, Kaunas, and Šiauliai region but by each age group (represented by grades 6–10, respectively) separately. In our study, the inclusion criteria were related to demographic characteristics (we included middle and late adolescents aged 15–18 years, both male and female) and the residence of the participants (we included the three largest cities in the country of Lithuania), and exclusion criteria were also related to demographic characteristics (the age of adolescents could affect the ability of the participants to answer the questionnaire correctly, as it was developed particularly for middle and late adolescent ages) and the residence of the participants (we excluded the district countries of Lithuania, as the main goal of this study was to evaluate the largest cities and to represent the overall situation in these parts of Lithuania.

Table 1 presents characteristics of the students in all variables as means and standard deviations (SDs) with *p* values and Cohen’s *d* scores. Statistically significant differences were found among boys and girls in terms of physical activity, father attachment, peer attachment, father trust, peer trust, mother communication, father communication, peer communication, father alienation, and peer alienation. Differences between 15–16- and 17–18-year-old students were statistically significant in terms of physical activity and peer alienation. Statistically significant differences were also found between two SES groups for physical activity, mother attachment, father attachment, parent attachment, mother trust, father trust, parent trust, father communication, parent communication, mother alienation, father alienation, and parent alienation.

The results show that males are physically more active than females (Cohen’s *d* = 0.27), younger 15–16-year-old students are more active than older adolescents aged 17–18 years (Cohen’s *d* = 0.21), and students of higher than average SES are more active than those of average or lower SES (Cohen’s *d* = 0.22).

Analyzing father/mother/peer attachment, parent/peer trust, communication and alienation domains, it was found that boys scored significantly higher than girls did on the father attachment component (Cohen’s *d* = 0.19). Females scored significantly higher than males did on peer attachment (Cohen’s *d* = 0.57), peer trust (Cohen’s *d* = 0.49), mother communication (Cohen’s *d* = 0.23), peer communication (Cohen’s *d* = 0.67), and father alienation (Cohen’s *d* = 0.19). The peer alienation score was significantly higher among males compared to females (Cohen’s *d* = 0.22). The same is true for 15–16-year-old teenagers compared to those of 17–18 years (Cohen’s *d* = −0.16).

Physical activity scores were higher in the group with an SES named as “above the average” compared to those who reported an “average or lower” SES (Cohen’s *d* = −0.22). Students from the SES group of “above the average” scored higher compared to those whose SES was “average or lower” on the following components: mother attachment (Cohen’s *d* = −0.17), father attachment (Cohen’s *d* = −0.27), mother trust (Cohen’s *d* = −0.18), father trust (Cohen’s *d* = −0.25), and father communication (Cohen’s *d* = −0.23). They scored lower on mother alienation (Cohen’s *d* = 0.16) and father alienation (Cohen’s *d* = −0.25).

Table 2 presents the potential interrelationships among physical activity and other variables studied. Physical activity had statistically significant positive correlations with all father and mother IPPA domains, but the strongest correlations were found with father attachment (r = 0.18, *p* = 0.01) (see Figure 1) father trust (r = 0.17, *p* = 0.01), and father communication (r = 0.19, *p* = 0.01) (see Figure 2), and negative correlations were found with father alienation (r = −0.13). Physical activity was also positively related to SES (r = 0.12) and negatively related to age (r = −0.13). No significant correlations were found between physical activity and peer trust, peer communication, or peer alienation in the general group. However, among boys, physical activity was positively related to peer trust (r = 0.14, *p* = 0.01) and peer communication (r = 0.13, *p* = 0.01). We found a positive relationship between physical activity and SES (r = 0.12, *p* = 0.01).

Table 3 presents multiple regression analysis. Regression analysis was conducted to examine the potential effect of studied variables on physical activity (Table 3). In order to find out which of the factors significantly contribute to physical activity, mother and father attachment domains, SES, age, and sex were included, as they showed significant relationships with physical activity in the previous analysis. Four of these variables had a statistically significant impact on physical activity, and the overall model explains 6.8% of the physical activity variation. More physically active adolescents scored higher on the father communication scale (Figure 2), and they were younger, male, and of higher SES; other variables were excluded from the model.

## 4. Discussion

This study aimed to examine links between adolescents’ engagement in physical activity and their attachment relationships with their mother, father, and peers. In the present study, we assessed whether the parent and peer attachment domains are associated with the level of physical activity among 15–18-year-old students from mainstream schools in Lithuania.

While investigating adolescents’ attachment to parents, we found that boys scored significantly higher than girls did in father attachment, father trust, and father communication domains. Females scored significantly higher than males did on mother communication. Girls scored higher than boys in the father alienation domain. The multiple regression analysis also revealed that father communication, adolescent age, gender, and SES are important aspects in relation to adolescents’ physical activity. The study results could be explained in relation to a tendency of a constant and persistent mothering role in the family, as communication and trust is present at most times [20]. Mothers demonstrate an emotional bond and warmth, while fathers demonstrate a motivational and empowering role [20]. In this sense, it might be that the father’s role is more important and more influential in enhancing adolescents’ physical activity, as the father is more associated with being an active role model. Researchers [20] have examined the parental role in the sports context and found that mother–adolescent interpersonal relationships are based on warmth while father–adolescent interpersonal relationships are based on support and involvement. The researchers emphasized that the parents’ roles are important in both age groups (15–16 years old and 17–18 years old). In the autonomy support range (scale), the mother’s role is stronger and more important for the younger group than in the group of 17–18-year-old athletes. Athletes with more experience (7 years or more) have higher father involvement than do players with less experience (6 years or less). With more experience in training and competition, basketball players have higher father involvement and support throughout their careers. Other researchers have found different findings. For example, the researchers [11] have found an important mothers’ role in increasing physical activity in children. Researchers [11] noted that moderate-to-vigorous physical activity and sedentary time in the family context was much stronger when both mothers and children were both at home together than when one or both were not at home. The researchers [11] emphasized and highlighted the importance in developing home-based family intervention strategies for increasing children physical activity.

It should be mentioned that such different findings could be influenced by many other factors. One reason could be the attachment formed in early childhood. In [11], parental involvement was evaluated, but attachment to parents was not. In this sense, it is difficult to evaluate such findings and to compare. Culture differences and different family traditions in the country could also play a role.

Moreover, in relation to adolescent–parent attachment, our study found statistically significant differences among two SES groups. All father and mother attachment, trust, and father communication scores were higher, and father/mother alienation scores lower, in the group with an SES named as “above the average” compared to those who reported “average or lower.” Family SES plays a key role in relation to the extra-curricular physical activity of students [21]. Moreover, the results of [18] support the idea that there is an association between SES and physical activity among adolescents. Girls with a higher SES are more physically active than those with a lower SES. In [21], the association between physical activity and age, sex, and SES components was examined, and it was found that age, sex, SES, mother and sibling physical activity, and peer influence were significantly associated with adolescent physical activity. The authors of [22] showed that perceived sports activity status of the mother was a significant factor of physical activity in boys, and perceived sports activity status of both parents (positive) was a significant factor for girls. Numerous studies have explored family and school influence, and parents and peers influence adolescents’ physical activity in different settings, but these links are not well understood [23,24,25]. The authors of [23] state that parental influences such as parents’ physical activity participation may play an important role in affecting youths’ physical activity, and results suggest that environmental and family-based interventions that increase fathers’ physical activity may help improve youths’ physical activity.

As the literature shows that peer influence in adolescence also has a significant influence on adolescent personality development [14], we assessed and compared parent and peer attachment in relation to adolescent physical activity. Our study results showed that peer trust had a weak positive correlation with physical activity only in boys. Females scored significantly higher than males did on peer attachment, peer trust, and peer communication components. Peer alienation score was significantly higher among boys compared to girls and among 15–16-year-old teenagers compared to those of 17–18 years. The study results demonstrate that males are more attached to their fathers, and female are more attached to their peers. In addition, the alienation scale also shows that younger males are more alienated from peers compared to older adolescents. Females had significantly lower scores on peer alienation than did males. The study results illustrate that, for females, it is important to maintain a stronger relationship with peers. These results could be explained by a greater wish to express oneself in a peer group compared with males. The study of [26] found similarly that girls are more attached to peers, and boys scored higher on all parent attachment subscales. It should be mentioned that no significant correlations were found between physical activity and peer trust, peer communication, and peer alienation in the general group.

In summary, we found that males, younger students, and those of higher than average SES were more physically active compared to females, senior adolescents, and those of average or lower SES. The study results illustrate that family is an important predictor in promoting adolescents’ participation in physical activity. There is a need to strengthen parent–adolescent and adolescent–peer interpersonal relationships so as to promote adolescent physical activity. These results also suggest that interventions should focus on adolescent girls who face a higher risk of inactivity. Interventions are also needed to involve all family members and promote educational interactions between them in order to increase physical activity.

## 5. Limitations and Directions for Future Research

Adolescence is recognized a critical stage of age in any context, and this study presents new insight on adolescent physical activity in relation to parent and peer attachment. The study highlights that parents, especially the father’s role, are of the highest importance in maximizing adolescent physical activity and suggests that family education regarding PA must be emphasized. However, study has limitations. Firstly, this study is not a longitudinal study. Adolescent–parent attachment at the early stage of their life was not known. Future studies could evaluate a long-term parent–adolescent attachment and the effect of such an attachment on the later development of close relationships with peers and on physical activity. Secondly, we included only middle and late adolescence stage of age. It would be important to include early adolescence stage of age and compare such periods in relation to parent–adolescent attachment and physical activity. Lastly, this study is a quantitative cross-sectional study. Future research should include similar evaluations by conducting qualitative study designs or mixed methods studies.

## 6. Conclusions

Physical activity is positively related to mother and father attachment as well as overall parent trust and parent communication, age, gender, and SES. Adolescent communication with one’s father, age, gender, and SES are the most important factors in relation to physical activity among 15–18-year-old adolescents.

## Figures and Tables

**Figure 1 ijerph-16-00866-f001:**
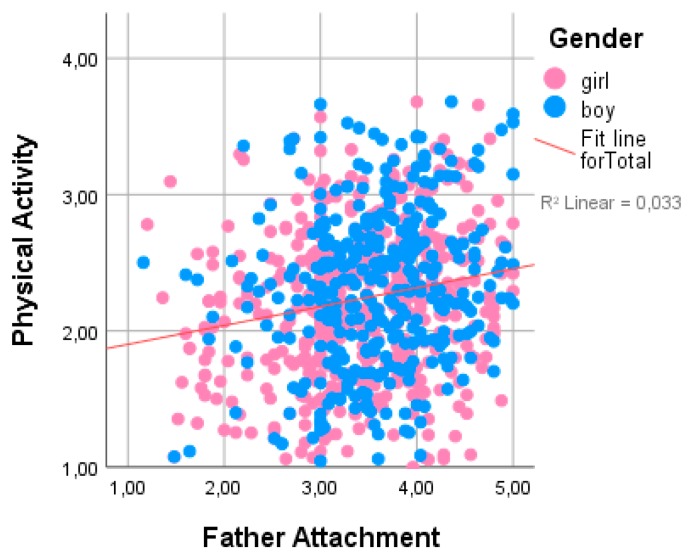
Interrelationships among PA, adolescent gender, and overall father attachment scale.

**Figure 2 ijerph-16-00866-f002:**
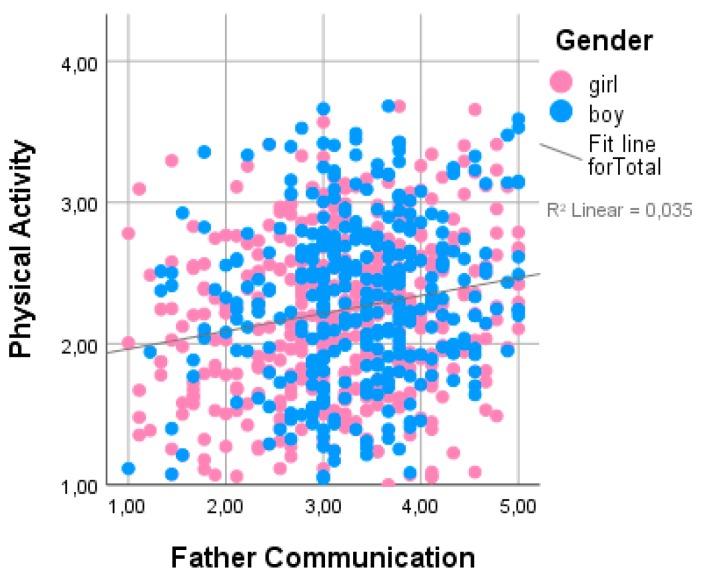
Interrelationships among PA, adolescent gender, and father communication subscale.

**Table 1 ijerph-16-00866-t001:** Mean scores (χ), standard deviation (*s*)***,*** and Cohen’s *d* effect size of physical activity (PA), mother/father/peer attachment, parent/peer trust/communication/alienation stratified by gender, age, and SES, *N* = 835.

Scale	Total (*N* = 835)		Girls (*N* = 480)		Boys (*N* = 355)		Cohen’s *d*	15–16 y.o. (*N* = 441)		17–18 y.o. (*N* = 394)		Cohen’s *d*	Average or Lower Status (*N* = 457)		Higher than Average Status (*N* = 378)		Cohen’s *d*
	χ	*s*	χ	*s*	χ	*s*		χ	*s*	χ	*s*		χ	*s*	χ	*s*	
Physical activity	2.2	0.6	2.2 ***	0.5	2.3 ***	0.6	0.27	2.3 **	0.6	2.2 **	0.6	0.21	2.2 **	0.6	2.3 **	0.6	0.22
Mother attachment	3.8	0.6	3.8	0.7	3.8	0.6	0.11	3.8	0.6	3.8	0.7	0.02	3.8 *	0.7	3.9 *	0.6	0.17
Father attachment	3.5	0.7	3.4 **	0.8	3.6 **	0.7	0.19	3.5	0.7	3.5	0.7	0.08	3.4 ***	0.7	3.6 ***	0.7	0.27
Parent attachment	3.7	0.6	3.7	0.6	3.7	0.5	0.00	3.7	0.6	3.7	0.6	0.05	3.6 ***	0.6	3.8 ***	0.6	0.26
Peer attachment	3.8	0.6	3.9 ***	0.6	3.6 ***	0.6	0.57	3.8	0.6	3.8	0.6	0.10	3.8	0.6	3.8	0.6	0.05
Mother trust	4.1	0.6	4.1	0.7	4.1	0.6	0.01	4.2	0.6	4.1	0.7	0.06	4.1 *	0.7	4.2 *	0.6	0.18
Father trust	3.8	0.8	3.7 *	0.8	3.9 *	0.7	0.17	3.8	0.8	3.8	0.8	0.08	3.7 ***	0.8	3.9 ***	0.7	0.25
Parent trust	4.0	0.6	3.9	0.6	4.0	0.5	0.11	4.0	0.6	3.9	0.6	0.08	3.9 ***	0.6	4.0 ***	0.5	0.27
Peer trust	4.1	0.7	4.2 ***	0.6	3.9 ***	0.7	0.49	4.1	0.7	4.1	0.7	0.04	4.1	0.7	4.1	0.7	0.01
Mother communication	3.6	0.7	3.7 **	0.8	3.6 **	0.7	0.23	3.7	0.7	3.6	0.8	0.04	3.6	0.8	3.7	0.7	0.12
Father communication	3.3	0.8	3.2 *	0.9	3.4 *	0.8	0.17	3.3	0.8	3.2	0.8	0.11	3.2 **	0.8	3.4 **	0.8	0.23
Parent communication	3.5	0.7	3.5	0.7	3.5	0.6	0.02	3.5	0.7	3.4	0.7	0.09	3.4 **	0.7	3.5 **	0.6	0.21
Peer communication	3.8	0.7	4.0 ***	0.7	3.6 ***	0.8	0.67	3.8	0.8	3.9	0.7	0.09	3.8	0.7	3.9	0.8	0.07
Mother alienation	2.4	0.8	2.4	0.8	2.5	0.7	0.03	2.5	0.7	2.4	0.8	0.08	2.5 *	0.8	2.4 *	0.8	0.16
Father alienation	2.6	0.8	2.7 **	0.8	2.5 **	0.8	0.19	2.6	0.8	2.6	0.8	0.01	2.7 ***	0.8	2.5 ***	0.8	0.25
Parent alienation	2.5	0.7	2.6	0.7	2.5	0.7	0.09	2.5	0.7	2.5	0.7	0.04	2.6 **	0.7	2.4 **	0.7	0.24
Peer alienation	2.6	0.6	2.5 **	0.6	2.7 **	0.6	0.22	2.6 *	0.6	2.5 *	0.6	0.16	2.6	0.6	2.6	0.7	0.05

* group differences are statistically significant at the 0.05 level; ** group differences are statistically significant at the 0.01 level; *** group differences are statistically significant at the 0.001 level; Cohen’s *d* effect size reference: 0.2 = small effect, 0.5 = medium effect, 0.8 = large effect according to Cohen (1988).

**Table 2 ijerph-16-00866-t002:** Correlations between physical activity and other study variables.

Scale/Variable	Physical Activity Score			
	Overall	Girls	Boys	
Mother attachment	0.088 *	0.082	0.118 *	0.01–0.19—very weak almost non-existent correlation
Father attachment	0.182 **	0.170 **	0.176 **	0.20–0.39 = weak correlation
Parent attachment	0.165 **	0.155 **	0.174 **	0.40–0.69 = medium correlation
Peer attachment	0.039	0.038	0.131 *	0.70–0.89 = strong correlation
Mother trust	0.095 **	0.074	0.131 *	0.90–1.00 = very strong correlation
Father trust	0.170 **	0.150 **	0.180 **	
Parent trust	0.165 **	0.142 **	0.189 **	
Peer trust	0.048	0.036	0.137 **	
Mother communication	0.076 *	0.084	0.105 *	
Father communication	0.187 **	0.192 **	0.158 **	
Parent communication	0.161 **	0.169 **	0.155 **	
Peer communication	0.035	0.042	0.127 *	
Mother alienation	−0.061	−0.065	−0.061	
Father alienation	−0.127 **	−0.113 *	−0.122 *	
Parent alienation	−0.111 **	−0.106 *	−0.106 *	
Peer alienation	−0.008	−0.017	−0.030	
Gender	0.132 **			
Age in years	−0.134 **	−0.195 **	–0.052	
SES	0.118 **	0.135 **	0.068	

Note: * Correlation is significant at the 0.05 level (2-tailed). ** Correlation is significant at the 0.01 level (2-tailed).

**Table 3 ijerph-16-00866-t003:** Regression analysis coefficients with PA as a dependent variable.

Model *		Unstandardized Coefficients		Standardized Coefficients	*t*	*p*
		B	Std. Error	Beta		
1	(Constant)	2.554	0.297		8.602	0.000
	Gender	0.123	0.038	0.108	3.204	0.001
	Age in years	−0.058	0.016	−0.118	−3.508	0.000
	SES	0.081	0.038	0.071	2.101	0.036
	Father communication	0.109	0.023	0.162	4.778	0.000

Note. * Regression with Enter method, model is significant (*p* = 0.000), R = 0.261, R_square_ = 0.068.

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
