# Peer review of "Links between Adolescents’ Engagement in Physical Activity and Their Attachment to Mothers, Fathers, and Peers"

_ijerph, 2019, doi:10.3390/ijerph16050866_

Round 1

Reviewer 1 Report

Dear authors

Thank you for your manuscript. Since high level of physical activity is a desirable aim at any age, but probably critical at adolescence, I am glad to review this article. I have some concerns regarding several points of the manuscript that I think could improve the final version.

INTRODUCTION

I wonder if it could be less repetitive, at some point it seems that the same information appears several times.

MATERIAL AND METHODS

Would it be possible to describe inclusion and exclusion criteria, and let the description of participants for the RESULTS section?

Page 3, line 103, add the IPPA abbreviation after the inventory name, as it appears here for the first time.

Data analysis, please clarify how every one of the tests was used, what was analyzed with each test

RESULTS

Please, consider presenting just one decimal on table 1

Table 1, consider deleting the p column and just adding the symbols

Table 2, results of physical activity correlation seem quite low. Please add a legend regarding what is it considered a high-moderate-low correlation level.

Table 3, please, consider changing the way information is presented because in my opinion, it is not clear the type of table presentation you have chosen for a regression analysis

DISCUSSION

Lines 190-191, I wonder it is a mistake to talk as peer relationships as predictors since correlation was not significant and they were not included in the regression model

Author Response

Dear Reviewer,

Thank you very much for the opportunity to revise and to resubmit the manuscript. We have submitted a substantially revised version of the manuscript. The revised manuscript and the response to reviewers comments could be found in the attached file. 

We appreciate the effort reviewers that we believe will strengthen this manuscript. 

Thank you.

Reviewer 2 Report

Dear authors,

I have read your paper with great interest. Your amount and quality of data can contribute a significant amount to the research of childhood physical activity. However, this paper needs still some major revisions. Please find attached my suggestions to improve this paper.

Best wishes and good luck.

Peer review for

„Links between adolescents' engagement in physical activity and their attachment to mothers, fathers, and peers”

l. 13

The following link is missing:   Importance of PA – Not enough meeting guidelines – support by parents might help to increase PA – therefore,   we examine the link between PA and attachment.

l.11

Grammar: offers (further grammar mistakes   were discovered, which are not commented further). Be also a bit more   creative with linkage words (e.g. l.184 ff: you started your sentences a lot   with “However,…”)

l.10

If you introduce abbreviations,   I suggest to make use of them in the following text

l.16

More information about the   population would be nice: where was that done? What is the proportion between   males and females?

l.17

Significantly related:  could you insert the statistical values   here?

l.22

“have a positive impact”: from the methods part in the abstract, I   understood that this is a correlational study design. Causal inference is   thus not possible.

l.24-25

Usually, abbreviations are not   part of keywords. Questionnaires neither, except if this questionnaire plays   a crucial role in your paper (in case it also validates the scale, for   example). Furthermore, I would put attachment or something related to your   main variable here.

l.28-30

I would suggest to put the   references about the consequences of not being sufficiently active after the   first sentence. And the references about the guidelines and adherence you can   leave here. This also applies for: l.38-44: there is no reference at all.   Reference [8] Dissertation: are there specific pages that you refer to? I   suggest to check for correct referencing further down in your paper.

l.45

it is not clear whether you are   talking about adolescents or children. I suggest to always stick to the same   term to not confuse the reader.

l.45-49

I would merge this paragraph   into one sentence, as introductory sentence for the following paragraph

l.60

What exactly is the difference   between parental involvement and attachment? Can you introduce the concept of   attachment? What might be the role of the peers (the peers are almost never   mentioned in the paper, that might need to be checked throughout the paper)

l.85

Is there an explanation for why   the age was split up into age group?

l.87

Which cities exactly are you   talking about? What were the inclusion criteria, what were the exclusion   criteria? Did you have ethical approval to conduct this study?

l.95ff

How do you know that the   questionnaire is appropriate for this age group? Is there a paper validating   this? Why did you use this questionnaire? Does it have high validity and   reliability values?

l.104

What means “IPPA”? References   missing in all paragraphs of “2.3. Measures”. Can you also give example-items   to make also concepts, such as peer alienation, clearer to the reader?

l.116ff

Did you come up with those   questions yourself? I am sure there is a background for this?

l.124f.

“The analyses included…”: What   outcome did you use which analysis for?

Results

I suggest to check formatting of   statistical values and tables. E.g. some letters need to be in italic. It is   unclear to me why you report Cohens d for some results but not for others. I   suggest to be consistent here

l.158ff

You have a cross-sectional study   design. Therefore, you cannot speak in causal terms (see similar comment   about line 22). Of course later in your discussion, you can then argue that,   although this study is cross-sectional, a causal link can be assumed since   child-parent relationships are likely to develop earlier and longer than does   PA…(or something like that ;-) )

l.184ff

I think the first sentence here   is just a repetition of the descriptives and not really necessary to answer   your research question. I also suggest to split this section into several   sub-sections. The first section could be the part about the parent attachment   and how it links to findings of other literature (this is completely missing   so far) and how the results can be explained. Why exactly might the fathers   role be so important? The argumentation here is not very clear. Second part   could be the results about the peers.

l.228 ff

This paragraph needs to be   developed further or merged into the sections about other findings. In my   opinion it could go well as conclusion after the paragraphs about the   correlations.

Discussion

What are the strengths and   limitations of your study?

l.233

“predictor”. See comment about   l.22

Author Response

(The authors gave the same response as above.)

Round 2

Reviewer 1 Report

Dear authors

The abstract has improved with the new data. The introduction is now clearer.

Regarding the description of the sample, I consider the RESULTS section is where it should appear.

I would include at the data analysis section what is a high-medium-low effect size regarding Cohen’s d

Table 1, please check if calculations were right, it changes quite a lot the data from the first to the second version. 2.25 is 2.3, but no 2.2, please check for errors and correct accordingly

Correlations still seem quite low, why don’t you add scatterplot graphs for the main findings?

Author Response

Dear Reviewer,

Thank you for the opportunity to revise and resubmit the manuscript. Your comments and suggestions helped us to finalize the manuscript and present our study in a professional manner. We appreciate your time and the effort in reviewing this manuscript. If there will be a need to correct the manuscript again, we are ready to do it.

Thank you very much.

Reviewer 2 Report

Thanks for the revised version of your paper about physical activity and attachment. This version significantly improved compared to the last version. Yet, I still have a few minor suggestions.

This study was cross-sectional, so be careful with causal claims, e.g. "communication is an important factor determining physical activity. 

The line of arguments is sometimes not very clear. I assume this comes with some language mistakes. I recommend to have read this paper by an external, native person to ensure a smooth reading flow.

l. 103: "The sample size was calculated basing on the number of pupils in the population" This is not clear

Author Response

(The authors gave the same response as above.)
